behaviour, evolution, ecology

CT scanning, 3D printing, niche construction, nest site selection, Lake Tanganyika, cichlid

**Authors for correspondence:**
Aneesh P. H. Bose
e-mail: abose@ab.mpg.de
Alex Jordan
e-mail: ajordan@ab.mpg.de

†Indicates co-first authorship.

# Structural manipulations of a shelter resource reveal underlying preference functions in a shell-dwelling cichlid fish

Aneesh P. H. Bose[1,2,3,†], Johannes Windorfer[1,2,3,†], Alex Böhm[1,2,3], Fabrizia Ronco[4], Adrian Indermaur[4], Walter Salzburger[4] and Alex Jordan[1,2,3]

[1]Department of Collective Behaviour, Max Planck Institute of Animal Behavior, Konstanz, Germany
[2]Centre for the Advanced Study of Collective Behaviour, and [3]Department of Biology, University of Konstanz, Konstanz, Germany
[4]Zoological Institute, University of Basel, Basel, Switzerland

APHB, 0000-0001-5716-0097; FR, 0000-0003-1583-8108; WS, 0000-0002-9988-1674; AJ, 0000-0001-6131-9734

Many animals can modify the environments in which they live, thereby changing the selection pressures they experience. A common example of such niche construction is the use, creation or modification of environmental resources for use as nests or shelters. Because these resources often have correlated structural elements, it can be difficult to disentangle the relative contribution of these elements to resource choice, and the preference functions underlying niche-construction behaviour remain hidden. Here, we present an experimental paradigm that uses 3D scanning, modelling and printing to create replicas of structures that differ with respect to key structural attributes. We show that a niche-constructing, shell-dwelling cichlid fish, *Neolamprologus multifasciatus*, has strong open-ended preference functions for exaggerated shell replicas. Fish preferred shells that were fully intact and either enlarged, lengthened or had widened apertures. Shell intactness was the most important structural attribute, followed by shell length, then aperture width. We disentangle the relative roles of different shell attributes, which are tightly correlated in the wild, but nevertheless differentially influence shelter choice and therefore niche construction in this species. We highlight the broad utility of our approach when compared with more traditional methods (e.g. two-choice tasks) for studying animal decision-making in a range of contexts.

## 1. Introduction

Animals often modify their environments in non-random ways [1–3]. They may choose among, construct or modify structures found in the wild in order to take shelter from predators or environmental extremes, increase foraging opportunities, enhance mate attraction potential and/or obtain space to rear offspring. Choosing and acquiring such structures represents a fundamental component of niche construction—the process by which organisms change selection pressures acting on them by modifying their biotic and abiotic environments [4]. Ultimately, resource selection and modification behaviours are adaptive if they increase fitness by better 'matching' an organism's environment to its phenotype [2,3]. Animals often have to decide among a suite of pre-existing structures or environmental elements that are either to be used or rejected. Bowerbirds, for example, gather ornaments of certain colours while removing others when decorating their bowers [5]. In other cases, after a structure has been chosen, certain attributes of the structure itself are modified. Hermit crabs, for example, alter the shells they reside in by hollowing them out [6,7], thereby increasing the fit between the environment and the organism. Often, the structures that wild animals must choose from are complex as they vary across many different axes including

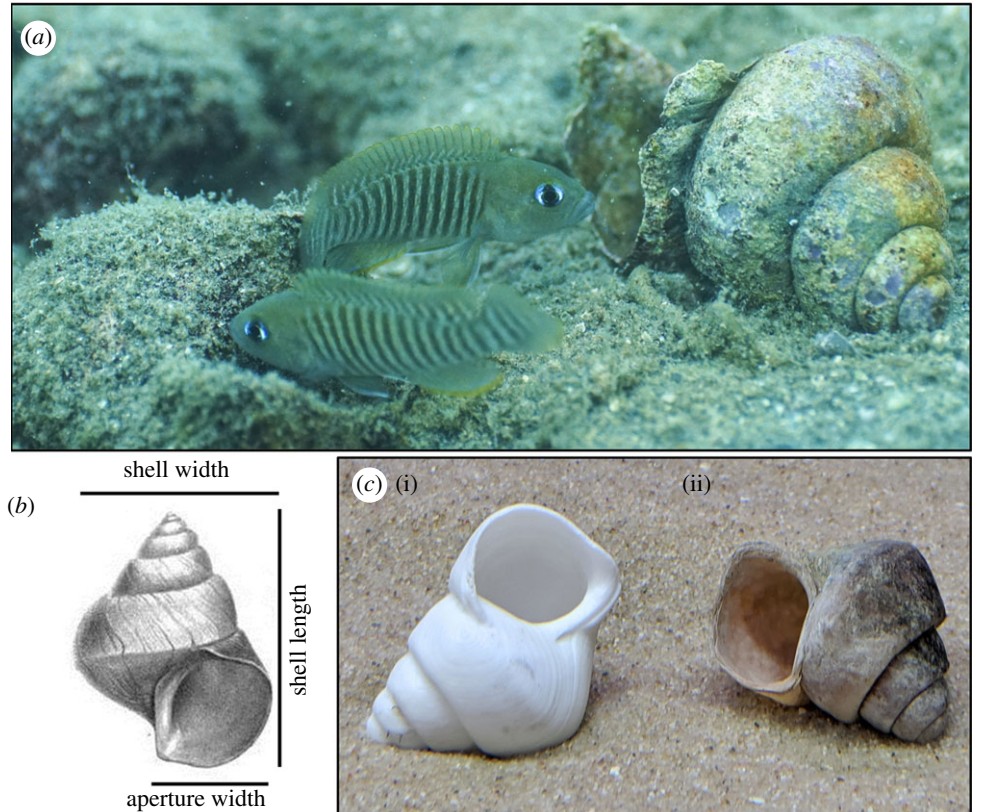

**Figure 1.** (*a*) Two male *Neolamprologus multifasciatus* interacting by a *Neothauma tanganyicense* shell in the wild ( photo credit: Jakob Guebel). (*b*) Sketch of a *N. tanganyicense* shell [12] indicating several axes of structural variation that were experimentally manipulated in this study. (*c*) A 3D-printed shell replica used in this study (i) beside a similarly sized natural shell (ii). (Online version in colour.)

size, shape, colour and texture [8]. Structural attributes are often highly correlated making it difficult to determine which attributes are most important for animal decision-making, and which are linked to changes in the selection regimes that animals experience.

The correlated nature of many structural attributes of the resources that animals choose among in the wild leads to a number of empirical difficulties. An animal choosing among a set of pre-existing shelters and opting for the largest one may, for example, receive a structure that simultaneously has a large entrance, more space to rear offspring, and more options to hide from predators. Similarly, an individual actively modifying a structure may create an end product that differs from the original one in more than one attribute. In these cases, it is difficult to disentangle what is being actively modified by niche construction behaviours, and which changes are simply by-products of the modifications. These problems can be further compounded if there is a mismatch between the perceived sensory environment of the study organism and that of the researcher, for example a colour that is obvious to the researchers but outside the visual sensitivity of study organism, leading to erroneous conclusions about 'adaptive' behaviour in the context of resource choice and niche construction [9]. Our ability to understand the evolutionary underpinnings of resource choice, and niche construction behaviour in general, will remain limited as long as we are unable to determine which elements of the environment are being chosen between or changed to fit the animal's needs. That is, we require a mechanistic knowledge of both the environmental inputs into the decision process as well as the fitness consequences of the behavioural outputs. One powerful approach to this problem is to experimentally

manipulate structural attributes of resources independently of one another and to test whether such manipulations alter the perceived value of the resource (e.g. [10,11]). Unfortunately, such highly controlled experiments are logistically demanding and rare.

In this study, we actively manipulate attributes of a commonly found structure in the environment of a social cichlid fish. We use a small shell-dwelling fish, *Neolamprologus multifasciatus*, endemic to Lake Tanganyika, East Africa, which facultatively uses empty *Neothauma tanganyicense* snail shells as brood chambers and as shelters to avoid predation (figure 1*a*) [13]. Where empty shells accumulate on the lake floor, *N. multifasciatus* exclusively use these shells in preference to all other shelter types. Because these shells are typically buried in sediment, *N. multifasciatus* dig shells out from the substrata and excavate sand from inside them, thereby creating shell-filled depressions on the lake floor separated by ridges of sand. Doing so creates large 'shell beds' containing thousands upon thousands of uncovered empty shells, which can stretch for hundreds of metres. The choice of which shells to occupy represents the first step in the process of niche construction at the individual level, producing shelters that individual *N. multifasciatus* must actively excavate and maintain, and also at the community level, creating structured shell beds on which numerous other species live—this is why the digging behaviour of shell-dwelling cichlids in general has also been referred to as 'ecosystem engineering' [14]. The environmental modifications associated with shell digging have clear fitness consequences for the excavating fish and may also have secondary effects when larger heterospecifics take the shells from smaller shell-dwelling *N. multifasciatus* [15], suggesting that larger species are too big to clear out shells of internal

sediment themselves, and need to forcefully take shells from smaller 'producer' species.

The *N. tanganyicense* shells excavated by *N. multifasciatus* can vary in overall size but have a highly stereotyped design (figure 1*b*), meaning that many structural attributes of these shells (e.g. shell length, shell width and aperture size) are highly correlated [13]. Specific structural attributes of the shells may convey different costs and benefits to the shell chooser. For instance, the size of a shell's aperture (i.e. shelter entranceway) could dictate how well the internal environment can be oxygenated by parental caregiving females tending to eggs laid on the inner walls of the shell whorls (shelter entrance size has been shown to mediate an important trade-off in the sand goby *Pomatoschistus minutus* [16]). Shell length, on the other hand, could determine the distance that an individual can retreat into the shell to avoid predators. Teasing apart the relative importance of these attributes, however, is often beyond the scope of small-scale observational studies, especially when structural attributes are tightly correlated, and so controlled experiments are a powerful way to explore these processes.

Our study presents a new paradigm for experimentally disentangling the often-correlated attributes of animal structures. This paradigm permits researchers to independently assess the relative contributions of different structural attributes that underlie animal resource-use decisions, and in our case shell choice, which is a conspicuous component of niche-constructing in *N. multifasciatus*. We used 3D scanning, modelling and printing to create accurate replicas of *N. tanganyicense* shells (figure 1*c*) wherein we exaggerate or diminish certain features. In particular, we manipulated overall shell size, shell length, aperture width and shell intactness. Doing so gives us insight not only into whether *N. multifasciatus* individuals have preferences for certain shells, but also what structural attributes (e.g. shell length, aperture size) drive those preferences. We also manipulated shell chirality, a highly conserved trait in natural conditions, opening the possibility to explore the evolution of lateral behaviour in fish as a consequence of shell morphology. We employed preference functions, a relatively novel analytical tool in behavioural ecology, to interrogate which attributes mediate shell choice. Our predictions were guided by the recognition that niche-constructing behaviours may be flexibly expressed across a lifetime or depend on environmental context or individual need [4]. If the tested attribute is unrelated to the decision to occupy the shell, then preference functions are expected to be flat and undirected, indicating no strong preference for a particular attribute. However, the functions could be curved, indicating preferences for specific attributes, and these may be categorized according to whether the functions are closed or open-ended [17]. Here, shell preferences should be for structural attributes that best 'fit' the phenotype of the chooser. For example, individuals of different sizes or sexes may choose shells according to attributes that best suit their needs.

## 2. Methods

### (a) Study species
*Neolamprologus multifasciatus* is one of the smallest cichlid species in Lake Tanganyika [18] and forms stable social groups consisting of up to approximately 20 individuals, with one to three adult males, up to five adult females, and the rest being juveniles and immature offspring [15,19]. *Neolamprologus multifasciatus* are mostly found living on large shell beds, made almost exclusively of empty *N. tanganyicense* snail shells that they use as both shelters from predators and as brood chambers for females to lay their eggs and tend to their larvae. Each group's territory contains at least as many shells as individuals, with each individual living in its own shell and returning to the shell routinely for maintenance [13]. While male and female *N. multifasciatus* may grow up to 45 mm and 35 mm, respectively, in captivity, they rarely grow past 30 mm and 21 mm, respectively, in the wild [13].

### (b) Shell structure manipulations
The shape of *N. tanganyicense* shells is highly stereotyped, meaning that as the (snail) shells grow, their structural attributes (e.g. shell length, shell width, aperture width) grow in or near isometry [13]. We measured the dimensions of a random sample of 113 *N. tanganyicense* shells, all collected at Chikonde Bay, Lake Tanganyika, Zambia (8°42′49.4″ S, 31°07′23.0″ E). Using ordinary least-squares regression, we tested for isometric growth between shell length, shell width, and aperture width (all log-transformed), and in no case did growth deviate significantly from isometry (95% CIs all included 1). Furthermore, a principal component analysis was able to reduce shell length, shell width and aperture width (all scaled) from the 113 shells into one component that explained 90.6% of the total variance. We then chose a representative, fully intact shell and CT scanned it (Bruker Skyscan 1174v2). We created a 3D model representation of the shell in which we could exaggerate or diminish certain attributes (using the software Autodesk Fusion360) and then 3D printed the resulting shell replicas using a custom-built printer (using PETG-Filaments; figure 1*c*). See electronic supplementary materials for a detailed explanation of scanning, modelling and printing the experimental *N. tanganyicense* shell replicas.

First, we manipulated overall shell size, expanding and shrinking it to cover the full range of shell sizes observed in the wild (see below). Next, we held shell size constant, and exaggerated or diminished three key structural attributes of the shells: shell length, aperture width and intactness (i.e. the number of holes in the shell). We chose to manipulate these three attributes because of the putative fitness effects that they could mediate. For example, shell length can determine how far within a shell a fish can retreat from predators. Aperture width could influence how quickly fish can enter a shell, how accessible it is to egg predators, and the degree of water flow into the internal shell environment for egg oxygenation. Intactness conveys information about the structural rigidity of the shell and the ease by which the shell might be broken or entered by predators.

For shell size, shell length and aperture width, we chose seven sizes that represented −3, −2, −1, 0, +1, +2 and +3 standard deviations (s.d.) around the population mean (table 1). Shells within *N. multifasciatus* territories are often not fully intact and can be chipped or partially broken. To manipulate shell intactness, we modelled and printed shell replicas with holes in the outer wall and/or with the last 8 mm removed from the apex of the shell (simulating shells with broken tips, which occur commonly in the wild). For detailed measurements of the manipulations, see electronic supplementary material, table S1.

### (c) Shell choice tasks and experimental setup
We used 40 male (mean ± s.d. = 36.5 ± 4.5 mm, standard length) and 40 female (30.1 ± 2.4 mm) adult and sexually mature *N. multifasciatus* in this experiment, which were F1–F3 descendants of wild stocks collected at Chikonde Bay (see electronic supplementary material for housing conditions). Ten males and 10 females were randomly assigned to one of four choice tasks, in which they were offered shells that varied in either overall

**Table 1.** Summary of size measurements (in millimetre) of 113 *Neothauma tanganyicense* snail shells collected at random from the wild in Chikonde Bay, Lake Tanganyika, Zambia.

| | minimum | 1st quartile | median | 3rd quartile | maximum | mean | s.d. |
|---|---|---|---|---|---|---|---|
| shell length | 23.1 | 39.1 | 43.3 | 47.0 | 60.0 | 43.2 | 5.8 |
| shell width | 19.3 | 29.6 | 32.1 | 35.4 | 41.6 | 32.3 | 3.8 |
| aperture width | 8.9 | 14.1 | 15.0 | 15.8 | 18.2 | 15.0 | 1.6 |

size, length, aperture width or intactness. In each choice task, the fish were placed in an experimental aquarium and presented with an array of seven 3D-printed shell replicas to choose from representing −3, −2, −1, 0, +1, +2 and +3 s.d. around the population means as described above (see electronic supplementary material, table S1). Body size did not differ significantly among choice tasks (generalized least-squares model, GLS, all $p > 0.05$), but did differ significantly between the sexes (males were larger than females, GLS, $p < 0.0001$). Each fish was tagged with a unique elastomer code (Northwest Marine Technology, Inc.) prior to the experiment. Experimental aquaria were either 50 × 75 × 35 cm (for the shell size, length and intactness choice tasks) or 35 × 75 × 35 cm (for the aperture width choice task) and had opaque walls to reduce external disturbance. Regardless of their dimensions, all tanks provided ample space for the shell-choosing fish, and tank dimensions did not vary within choice tasks. As *N. multifasciatus* are highly social fish, we conducted each shell choice task in a social setting to ensure normal behaviour and avoid isolation stress. To this end, each aquarium held a transparent acrylic cylinder in the centre (15 cm in diameter, 24.5 cm in height, perforated with 48 holes to allow water exchange), which permanently housed one male and two female *N. multifasciatus* individuals, each with a natural *N. tanganyicense* shell, to act as a social stimulus group. The space provided to the stimulus group within their cylinder reflected the area that would be occupied by a similarly sized group in the wild. We placed the seven shell replicas evenly-spaced around the acrylic cylinder. For each choice task, the experimental fish was introduced into the tank and allowed 20 h to choose one of the shell replicas. Each fish was given the same choice task four times with the exception of three fish, which were tested three times and one fish, which was tested five times, summing to 318 trials in total (fish were given at least four days in between trials). Note that omitting the fish that were not tested four times does not qualitatively change our conclusions. After each choice trial, one experimenter performed a spot observation from behind a blind (to avoid the fish seeing the experimenter) and recorded whether the fish had chosen a shell replica. This was determined based on whether the fish was hiding in, resting in front of, or swimming immediately above one of the replicas. The repeated observations of the fish choosing among the replicas were used to calculate preference functions for each fish (see below). In between trials, we levelled the bottom sand layer to hide any evidence of digging by the previous fish and randomized the order and position of the shells in the aquarium.

## (d) Preference functions

With each fish having been used in four replicate trials of their specific choice task, we used their shell choices to build preference functions using the program PFunc (v. 1.0.1) [17]. Preference functions have recently been adopted into the field of behavioural ecology and are largely used to study mate choice, i.e. the preferences that individuals show for different phenotypic traits of potential mates (e.g. [20–22]). At their core, preference functions are curves (splines) that are fitted to data indicating which values of a trait are more or less attractive to an observer [17]. In PFunc, these curves are fitted using generalized additive models. Certain measurements, so called preference function *traits*, can then be extracted from these curves that lend information about the shape of the curves, and hence the underlying preferences of the chooser (see electronic supplementary materials figure S1 for example). For our analyses, we extracted three preference function traits: (i) peak preference, which is the trait value (on the *x*-axis) where the function is at its maximum, indicating the most preferred trait value; (ii) preference strength, which is the steepness of the preference function slope as it drops away from the point of peak preference; and (iii) tolerance, which is the width of the preference function at a pre-specified height, describing the range of trait values over which the function remains relatively high (here, we measure tolerance at half of the peak height, using the 'broad' definition, see [17]). Collectively, these preference function traits convey information about which shell attributes are preferred, which value of an attribute is considered most attractive, and whether the chooser tolerates any deviation away from their most preferred values. For the shell size, shell length, aperture width and shell intactness tasks, we calculated preference functions for each fish individually, so long as they successfully chose a shell at least three times out of the four or five trials that they participated in. Ninety per cent of the focal fish (or 72 out of 80 fish) successfully chose a shell at least three times; in 10% of the trials (32 out of 318 trials), the fish did not make a clear choice.

## (e) Shell chirality-experiment

Virtually all *N. tanganyicense* shells have the same chirality (dextral; i.e. right-coiled) and so we also investigated whether chirality influenced shell choice. To achieve a sinistral form (i.e. left-coiled), we mirrored and 3D printed average *N. tanganyicense* shell models. Unlike previous structural attributes that we tested (i.e. shell length, aperture width), which vary along a continuous axis, shell chirality is binary and represents variation that would almost never be encountered in the wild. Chirality choice tasks were conducted using a random subset of 20 females from the original 40, though this was done only after all other choice tasks had been completed (mean ± s.d. = 30.4 ± 2.3 mm, standard length). Later, we also tested a set of 20 males randomly selected from our stock population (mean ± s.d. = 34.2 ± 4.6 mm). Here, each fish participated in only a single two-choice task. These choice tasks took place in the larger of the two experimental tank types described above, but involved two opposing shell options rather than seven.

## (f) Statistical analysis

### (i) Are observed preference function traits non-random?

All statistical analyses for this study were conducted in R (v. 3.6.3.) [23]. We first tested whether the choices that fish made in the shell size, shell length, aperture width and shell intactness choice tasks were non-random. To do this, we calculated the null expectations for our preference function traits under purely random choice. We simulated 1000 replicates of a

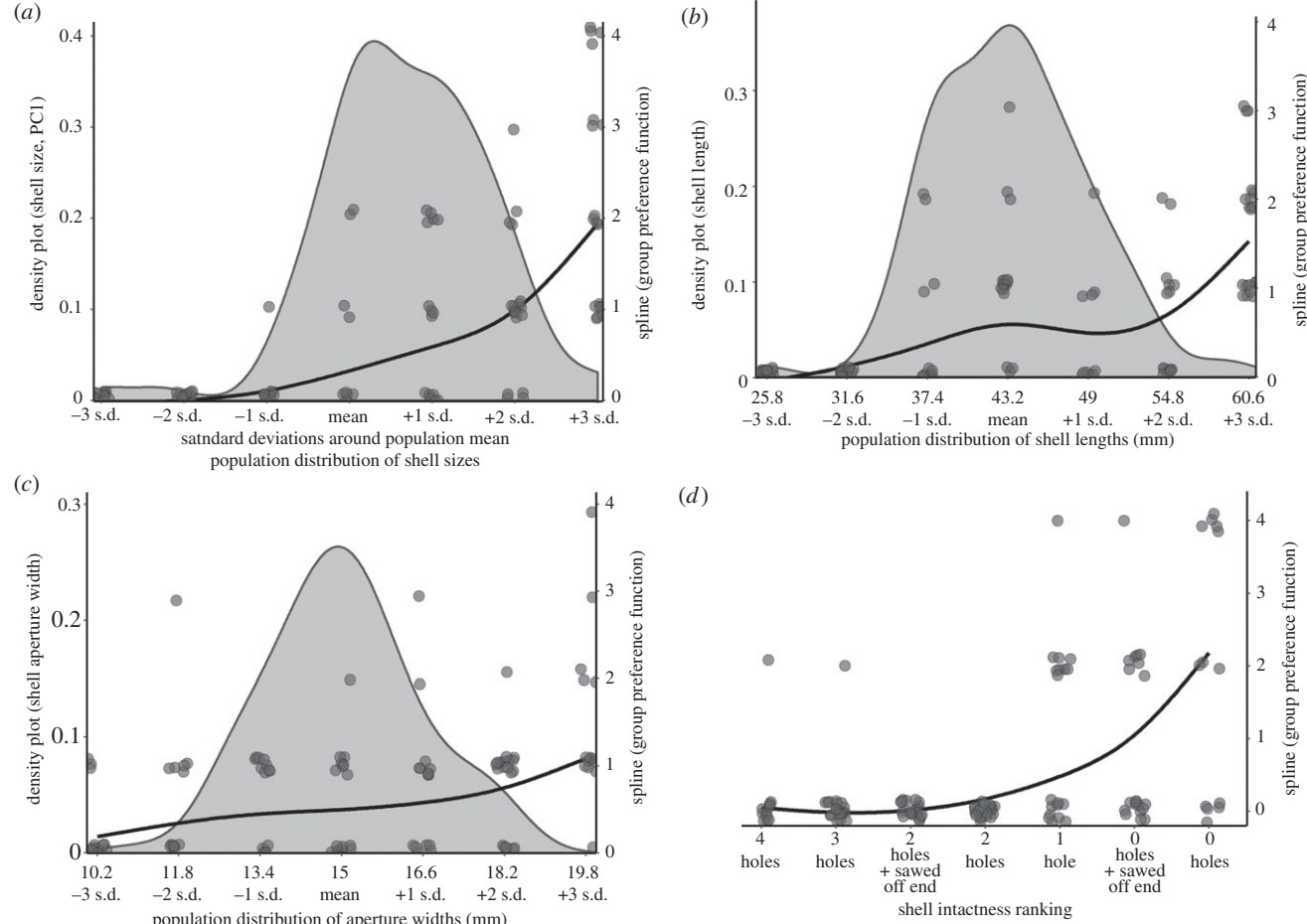

**Figure 2.** Results of shell choice tasks in which (*a*) overall shell size, (*b*) shell length, (*c*) shell aperture width, and (*d*) shell intactness were manipulated using *Neothauma tanganyicense* shell replicas as described in Methods. Grey density plots illustrate the natural distribution of these attributes as observed in the wild (*n* = 113 shells; see Methods). The population density plot for shell size was created by using a principal component analysis to reduce shell length, shell width, and aperture width data (all scaled) from the 113 collected shells into one composite variable, PC1 (accounting for 90.6% of the total variance). There are no comparable population data for shell intactness. The curves are spline fits that represent group-level preference functions for all male and female *N. multifasciatus* pooled together. Splines were generated in the program PFunc [17]. Each point represents the choices made by individual fish (i.e. the number of times individual fish chose shell replicas of a given form).

fish making four random choices out of seven shell replica options. We then calculated their respective preference function traits for each of these 1000 replicates. This mimicked what would happen if fish chose shells randomly during our trials. We then tested whether our observed preference function traits differed from those derived from random choice. To do this, we fitted a linear model (ln-transforming or using a generalized least-squares model when variances were heteroscedastic across groups), to each of the preference function traits (peak preference, preference strength and tolerance). In each model, we included choice task as a predictor variable (a 5-level categorical variable representing the shell size, shell length, aperture width and shell intactness choice tasks, plus the random choice). We then used Dunnett's contrasts to compare each choice task back to random choice while accounting for multiple comparisons (using the 'multcomp' R package, v. 1.4-8 [24]).

### (ii) Does shell structure, sex or body size influence preference function traits?

Next, we fitted additional linear models (or generalized least-squares models when variances were heteroscedastic across groups) to our preference function traits. We first included choice task as a predictor variable (a 4-level categorical variable representing the shell size, shell length, aperture width and shell intactness tasks). Then, we tested whether body size (continuous variable, mm) and sex (2-level categorical variable: male or

female), two strongly correlated variables, should also be included in the models together. Adult males are larger than adult females in *N. multifasciatus* and this was especially the case in our sample; based on a generalized least-squares model, the average standard length of our focal males, 36.5 mm, was significantly longer than that of our focal females, 30.1 mm (est. ± s.e. = 6.47 ± 0.81, $t_{78} = 8.04$, $p < 0.0001$). We used both variance inflation factors (VIFs, 'car' R package [25]) and likelihood ratio tests (LRTs) to examine the consequences of including both sex and body size in our models. VIFs suggested high levels of multicollinearity (e.g. values > 5), and LRTs suggested that sex did not improve model fits (all $p > 0.05$). We therefore included only body size in our models; however, we temper our interpretations of the results given the tight correlation between sex and size (also, see electronic supplementary material, figure S2 for a breakdown of figure 2 split by sex). All pairwise interactions between choice task and body size were tested but dropped if non-significant. Body size was mean-centred so that when interactions where significant, the model coefficients would be interpretable. We tested all pairwise contrasts between choice tasks accounting for multiple comparisons (using the 'multcomp' R package, v. 1.4-8 [24]).

### (iii) Is one shell chirality preferred over the other?

We tested whether the sexes differed in their likelihood of choosing the shell with the opposite chirality (i.e. left-coiled) by fitting a binary logistic generalized linear model (GLM) to their choices

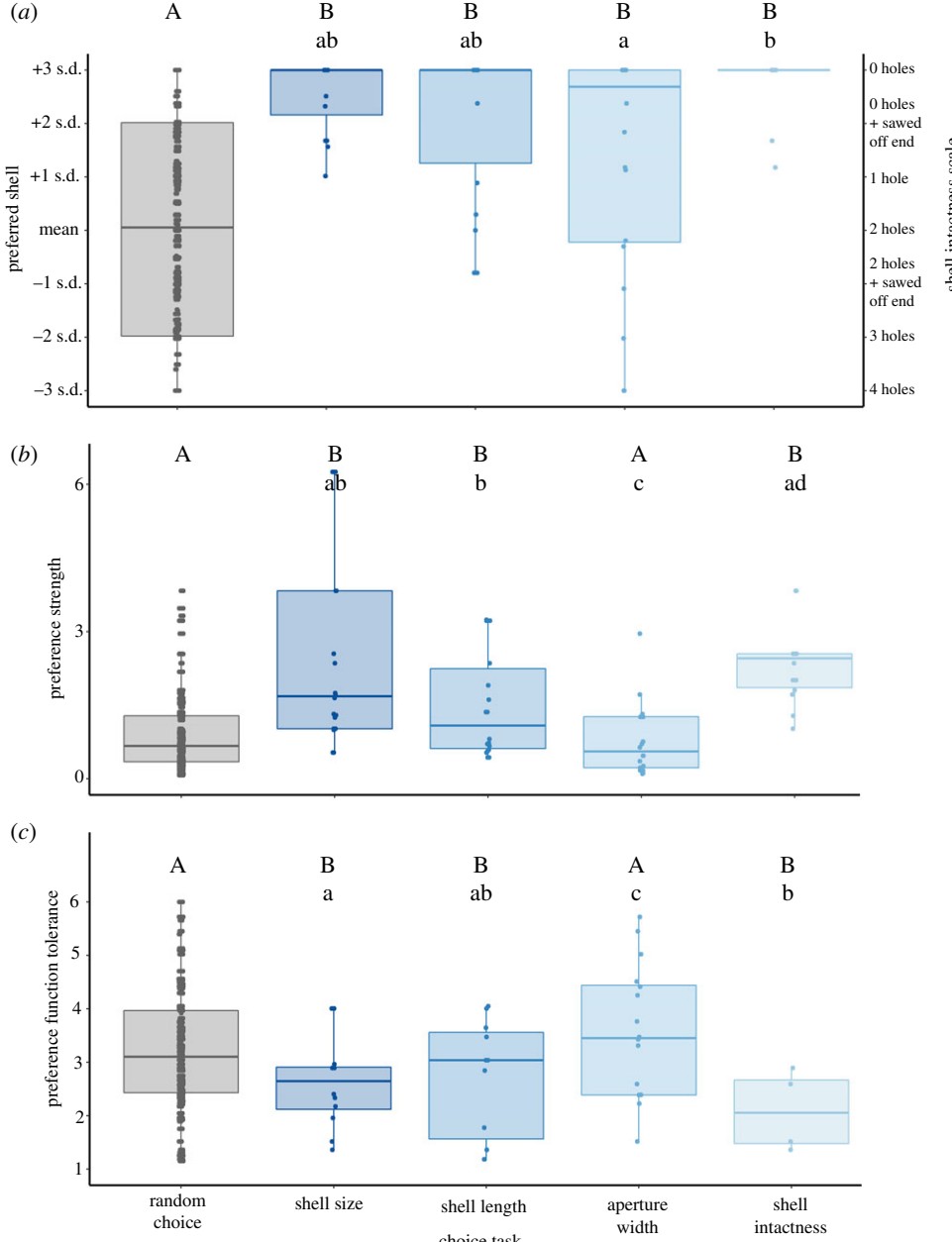

**Figure 3.** Preference function traits—(*a*) peak preference, (*b*) preference strength and (*c*) tolerance—for each of the experimental groups. The left-most group in each panel represents the preference function traits derived by simulating random choices. Each of the remaining groups represents a choice task where fish chose among 3D-printed shell replicas that varied with respect to one structural attribute, either overall shell size, shell length, aperture width or shell intactness. Note the *y*-axis on the right in (*a*), which indicates the ranking of shells with increasing degrees of intactness. Upper-case letters denote statistical differences between the choice trials and random conditions (Dunnett's contrasts), while lower-case letters denote pairwise differences among the choice trials themselves. (Online version in colour.)

and including sex in the model as a categorical predictor variable. We also tested whether *N. multifasciatus*, regardless of sex, preferred one shell chirality above the other by refitting the above model with only the intercept term, and then tested whether the intercept differed significantly from 0 (indicating no preference).

## 3. Results

### (a) Peak preference

Peak preference can be interpreted as the *most preferred value* for the structural attribute being experimentally varied. Mean (±s.d.) peak preferences were 2.6 ± 0.7 for shell size, 2.1 ± 1.5 for shell length, 1.4 ± 2.2 for aperture width and 2.8 ± 0.6 for shell intactness (units are in s.d. around the population mean; figure 2*a–d*). The peak preference for each structural attribute that we tested was significantly higher than the

expectations of random choice (GLS, all $p < 0.034$). The average peak preference for shell aperture width was significantly lower than that for shell intactness (GLS, est. ± s.e. = −1.56 ± 0.49, $z = −3.19$, $p = 0.0066$) and nearly so for shell size (est. ± s.e. = −1.20 ± 0.48, $z = −2.50$, $p = 0.053$; figure 3*a*). No other contrasts between structural attributes were significant. Overall, larger bodied fish showed stronger preferences for shells with more exaggerated attributes or intactness (GLS, est. ± s.e. = 0.07 ± 0.02, $t_{67} = 2.94$, $p = 0.0045$).

### (b) Preference strength

Preference strength can be interpreted as the steepness by which the preference function drops off with increasing deviation away from the point of peak preference [17]. Mean (±s.d.) preference strengths were 2.6 ± 2.1 for shell size, 1.5 ± 1.1 for shell length, 1.0 ± 1.4 for aperture width and

2.6 ± 1.3 for shell intactness (preference strength is a unitless measure [17]; figure 2a–d). Average preference strengths were significantly higher than expected by random choice for shell size (LM, ln transformation, est. ± s.e. = 1.07 ± 0.21, $t_{1067}$ = 5.18, $p < 0.0001$), shell intactness (est. ± s.e. = 1.27 ± 0.25, $t_{1067}$ = 5.16, $p < 0.0001$) and shell length (est. ± s.e. = 0.57 ± 0.22, $t_{1067}$ = 2.61, $p = 0.036$), but not for aperture width (est. ± s.e. = −0.15 ± 0.21, $t_{1067}$ = −0.75, $p = 0.91$). Average preference strength was higher for shell size than aperture width (LM, est. ± s.e. = 1.17 ± 0.24, $t_{64}$ = 4.97, $p < 0.001$) and higher for shell length than aperture width (est. ± s.e. = 0.82 ± 0.24, $t_{1067}$ = 3.37, $p = 0.0068$). Average preference strength for shell intactness was also higher than for shell length (est. ± s.e. = 0.74 ± 0.27, $t_{64}$ = 32.78, $p = 0.035$) and aperture width (est. ± s.e. = 1.56 ± 0.26, $t_{1067}$ = 5.96, $p < 0.001$; figure 3b). Larger-bodied fish also displayed higher preference strength scores (est. ± s.e. = 0.092 ± 0.21, $t_{1067}$ = 4.42, $p < 0.0001$).

### (c) Tolerance

Tolerance can be interpreted as the willingness of an individual to accept a *range* of values of the structural attribute being experimentally varied. Mean (±s.d.) tolerance values were 1.8 ± 1.2 for shell size, 1.8 ± 1.3 for shell length, 3.0 ± 1.6 for aperture width, and 1.2 ± 0.7 for shell intactness (units are in s.d.; figure 2a–d). Average tolerances for shell size (GLS, est. ± s.e. = −0.91 ± 0.26, $t_{1067}$ = −3.47, $p = 0.0021$), shell length (est. ± s.e. = −0.82 ± 0.31, $t_{1067}$ = −2.69, $p = 0.029$) and shell intactness (est. ± s.e. = −1.55 ± 0.19, $t_{1067}$ = −8.26, $p < 0.0001$) were significantly lower than the expectations of random choice. The average tolerance for shell aperture width (est. ± s.e. = 0.25 ± 0.35, $t_{1067}$ = 0.71, $p = 0.93$) was not significantly different from random. Fish showed the lowest tolerance for variation in shell intactness, preferring the most intact shells. Their average tolerance for shell intactness was lower than for shell size (GLS, est. ± s.e. = −0.89 ± 0.29, $t_{67}$ = −3.10, $p = 0.010$) and aperture width (est. ± s.e. = −1.98 ± 0.40, $t_{67}$ = −4.91, $p < 0.0001$; figure 3c). Tolerance for aperture width was higher than for the other choice tasks, as the experimental fish chose shells spanning many different aperture widths. Average tolerance for aperture width was higher than for shell size (est. ± s.e. = 1.08 ± 0.40, $t_{67}$ = 2.73, $p = 0.032$) or shell length (est. ± s.e. = 1.20 ± 0.44, $t_{67}$ = 2.73, $p = 0.031$; figure 3c). Lastly, larger bodied fish showed lower tolerances, i.e. they had more stringent preferences (GLS, est. ± s.e. = −0.13 ± 0.03, $t_{67}$ = −4.57, $p < 0.0001$).

### (d) Chirality

Females chose one of the two shell replicas in 20 out of their 20 trials, while males chose one in 14 out of their 20 trials. The sexes did not differ from each other in their chirality choices (GLM, est. ± s.e. = 0.41 ± 0.95, $z = 0.43$, $p = 0.67$). Overall, shells bearing the normal chirality (i.e. right-coiled) were preferred more often than shells bearing the opposing chirality (GLM, est. ± s.e. = 1.54 ± 0.45, $z = 3.42$, $p = 0.0006$). Sixteen out of 20 females and 12 out of 14 males chose the right-coiled shell replicas in their trials.

## 4. Discussion

Rather than passively responding to the conditions they find themselves in, many organisms exert an influence on their biotic and abiotic surroundings, thereby changing the selective regimes they experience [2–4,26]. A common form of such niche construction is the choice and/or modification of physical elements in the environment for use as nests or shelters. Yet because physical attributes of these structures can be tightly correlated with one another, the task of understanding which attributes underlie resource choice and hence niche construction behaviours is complicated. However, new techniques in 3D scanning and printing have created experimental opportunities for behavioural ecologists to overcome these challenges. Here, we employed these techniques to manipulate shelter structure and uncover the hidden preference functions that underlie resource choices in *N. multifasciatus*.

We observed non-random, open-ended preference functions for each structural attribute, jointly suggesting that *N. multifasciatus* have preferences for the most exaggerated or intact shell forms that we provided, particularly in the choice tasks where all shell replica options retained naturalistic dimensions and proportions (i.e. the shell size and intactness tasks). The preference function traits often differed across the structural attributes, implying that each attribute may be differentially related to the costs and benefits of shelter ownership. Preference function traits for shell size and intactness were consistently different from chance, while this was not the case for shell length or aperture width. Furthermore, preference function traits often implied weaker preferences for aperture width than for overall shell size and intactness, with preference for shell length being intermediate. Overall, this suggests that large, intact shells are highly valued, and that shell length may be more important than aperture width, at least for the range and combination of attributes that we tested. Fully intact shells are expected to be more valuable than broken shells, because holes in the shell walls can indicate structural frailty and may also serve as additional entranceways for predators and competitors. However, a recent survey conducted in the field wherein all *N. tanganyicense* shells were inspected from 24 *N. multifasciatus* territories suggests that only approximately 30% of shells are fully intact (A.P.H.B. & A.J. 2019, personal observations), which probably forces some fish to occupy broken shells when no other vacancies are available. Shell length may play a role in predator avoidance by allowing fish to hide further away from their shelter entranceways when approached by a predator. Anecdotally, predators such as *Mastacembelus* spp. or *Neolamprologus tetracanthus* are sometimes able to extract *N. multifasciatus* from their shells by either partially entering the shells with their heads or by creating a suction seal between their mouths and the shell aperture (A.P.H.B.& A.J. 2019, personal observations). The distance that a fish can retreat into its shell could therefore have strong survival benefits. Interestingly, aperture width appeared the least valued trait. This was surprising because nest entrance size is known to be an important structural attribute in other study systems. For example, in sand goby (*Pomatoschistus minutus*), nest entrance is at the centre of an important trade-off between defensibility of the nest and parental care [16,27]. In marsh tits (*Poecile palustris*), small nest entrances are important in preventing access by large predators [28,29]. Our data suggest that *N. multifasciatus*'s preference for large shells may be driven predominantly by a preference for long shells rather than for shells with wide apertures. Given the continuous narrowing in chamber width as fish retreat into the shell, it is possible that external aperture is less important, as eggs can be laid anywhere along the interior wall, in a location that

maximizes the trade-off between parental care and defence against predators. In a similar vein, females of the Lamprologine cichlid, *Julidochromis transcriptus*, deposit their eggs at particular locations along narrowing, wedge-shaped, rocky crevasses as a way of controlling male paternity and care patterns [30,31]. Future studies will manipulate additional structural attributes, including colour and surface texture of the shell replicas as these attributes are foreseeably linked to fitness outcomes through mimesis with the substrate or egg adherence.

Adult male *N. multifasciatus* are larger than females and so sex was correlated with body size in our study. Larger fish often showed stronger preferences than smaller fish (i.e. they had higher peak preferences, higher preference strengths and lower tolerances in favour of the most exaggerated shell forms). This was despite the fact that the largest males in our study should still have been physically able to enter the apertures of average-sized shell replicas presented to them. These results are consistent with a resource choice strategy that matches the resource to the phenotype of the chooser. We had initially predicted the sexes to differ with respect to their preferences, since males use *N. tanganyicense* shells primarily for shelter and females use the shells additionally as brood chambers. We, therefore, expected females to attend more to attributes such as aperture width or shell intactness, which could modulate the internal oxygen environment for egg rearing. Nevertheless, we found that regardless of body size (and presumably of sex), fish preferred the more exaggerated shell replicas. In the wild, where shells with extreme forms are rare and are likely to be monopolized by larger fish, females may therefore be less likely than males to secure their most preferred shells. Future experiments will be needed to investigate potential sex effects further.

Our most common preference function shape was openended, suggesting a preference for the most exaggerated shell replicas presented in this study. Our most exaggerated shell replicas, printed to possess attributes that were three standard deviations above the population mean, represent extreme but still ecologically plausible forms. Future studies may wish to expand the attributes even further to formally test preferences for supernormal stimuli. Preferences for supernormal stimuli may arise when, under ecologically realistic conditions, there are stronger adaptive benefits to preferring stimuli that are slightly exaggerated than there are costs to selecting stimuli that are too exaggerated [32,33]. Preferences for supernormal stimuli may also occur if the stimulus is exploiting a sensory bias in the chooser, irrespective of whether selection is acting directly on the preference [34]. The open-ended preferences that we uncovered can illustrate which structural attributes are important for shell choice, but they are unlikely to be fully realized under natural conditions through the lack of availability of shells with such extreme attributes. Furthermore, in contrast with the typical use of preference functions when studying sexually selected traits, here there are no co-evolutionary processes occurring that would result in an alteration to the trait(s) being preferred. The fish choose among the shells of dead snails and their preferences for any shell attributes cannot generate selection on those traits in the living molluscs.

All *N. tanganyicense* shells we observed in the wild bear the same right-coiled chirality, and yet the *N. multifasciatus* in our experiment were able to distinguish between the shell replicas of opposing chirality that were provided to them. On average, the fish favoured the natural, right-coiled form and there are two possible, non-mutually exclusive explanations for this preference. First, because the experimental fish had no opportunity to interact with shells of the opposite chirality prior to our choice task, our observations might be explained by an aversion to novel stimuli or habituation to experienced cues. Alternatively, the use of right-coiled shells over evolutionary time may have generated a lateralized behavioural response in shell-dwelling cichlids. Behavioural, morphological, and even neuroanatomical lateralization are well known in the scale-eating Tanganyikan cichlid *Perissodus microlepis* [35–38], and with our technique the possibility to explore lateralized behaviour and morphology more broadly in shell-dwelling cichlids offers an exciting new avenue of research in this area.

A valuable next step will be to take this experimental paradigm to the field or more ecologically realistic conditions, where the social environment, predation risk and reproductive or parental status vary more naturally. Under natural conditions, shell choices are more likely to face costly trade-offs as particular structural attributes could convey multiple ecological or social functions such that no single function can be optimized for (e.g. a Pareto front [39]), and this is likely to differ among individuals and environmental contexts. For example, fish may prioritize different structural attributes depending on whether they face high or low predation threats, reproductive or non-reproductive contexts, parental or non-parental care duties, etc., and no single shell choice may satisfy all these demands. Note, however, that while preferences for different attributes may shift with context, the manifestation of these preferences (i.e. shell choice) may not differ greatly in the wild, where the attributes of natural shells are tightly correlated and opportunity for choice limited.

Our results contribute to a wider understanding of animal decision-making, which requires the integration of information about alternative choices, each of which can differ with respect to various attributes. Nest-site choice and mate choice have both been popular topics in which to investigate the principles of decision-making and the use of multiple cues (mate choice: [40–42]; habitat and nest-site choice: [10,43]). Studies are particularly valuable when they manipulate or assess multiple attributes at once so as to disentangle their independent or interactive roles. For example, Franks *et al*. [10] investigated nest choice strategy in *Temnothorax albipennis* ant colonies by independently manipulating multiple nest attributes and showed that the ants ranked nest cavity lighting conditions above cavity height, which was itself ranked higher than nest entrance size. Bose *et al*. [44] used large-scale field data of breeding plainfin midshipman fish, *Porichthys notatus*, to statistically parse the relative importance of different characteristics of the male phenotype and nest structure on female choice and male reproductive success. Studies such as these either use hand-made nest models that differ dramatically from one another, precluding any fine-scale conclusions, or they require exceptionally large and detailed datasets. Our current study presents a new paradigm for conducting highly controlled studies of animal decision-making by highlighting the use of 3D scanning, modelling and printing in combination with preference functions as an analytical tool. Such techniques should be applicable to a wide variety of research questions on how animals evaluate and discriminate among complex alternatives.

Ethics. The work presented in this study was performed under the approval of the Tierschutzgesetzes (TierSchG) Baden-Württemberg, as given by permit no. G 18/75.

**Data accessibility.** All analyses in this study can be reproduced with data available from the Dryad Digital Repository: https://doi.org/10.5061/dryad.4b8gtht8s [45].

**Authors' contributions.** A.P.H.B., J.W. and A.J. devised the experiment with assistance from F.R., A.I. and W.S. F.R. CT scanned the shells. A.P.H.B. 3D modelled and printed the shell replicas. J.W. and A.P.H.B. conducted the experiment and analysed the data. A.P.H.B., J.W. and A.J. wrote the manuscript with input from all other co-authors.

**Competing interests.** We declare we have no competing interests.

**Funding.** A.P.H.B. was supported by an Alexander von Humboldt Research Fellowship. This research was funded by the Deutsche Forschungsgemeinschaft (DFG, German Research Foundation) under Germany's Excellence Strategy (EXC 2117—422037984). W.S. was supported by the Swiss National Science Foundation (grant no. 179039).

**Acknowledgements.** We thank members of the Jordan laboratory and the Department of Collective Behaviour for many fruitful discussions. Thank you to members of the Jordan and Kohda labs for assistance collecting *N. tanganyicense* shells from the wild. We also thank the Department of Fisheries in Mpulungu (Zambia), especially L. Makasa and T. Banda, for kindly supporting our research at Lake Tanganyika.

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
