## [Reviewer comments · Proceedings of the Royal Society B: Biological Sciences]

Review History

RSPB-2020-0127.R0 (Original submission)

Review form: Reviewer 1 (Rafael L. Rodriguez)

Recommendation

Accept with minor revision (please list in comments)

Scientific importance: Is the manuscript an original and important contribution to its field?

Excellent

General interest: Is the paper of sufficient general interest?

Excellent

Quality of the paper: Is the overall quality of the paper suitable?

Excellent

Is the length of the paper justified?

Yes

Should the paper be seen by a specialist statistical reviewer?

No

Do you have any concerns about statistical analyses in this paper? If so, please specify them explicitly in your report.

No

It is a condition of publication that authors make their supporting data, code and materials available - either as supplementary material or hosted in an external repository. Please rate, if applicable, the supporting data on the following criteria.

Is it accessible?

Yes

Is it clear?

Yes

Is it adequate?

Yes

Do you have any ethical concerns with this paper?

No

Comments to the Author

This manuscript tests for preferences for the features of the snail shells that a Lake Tanganyika cichlid fish uses as refuges. This may help understand which shell features are of importance for the cichlids' survival or reproduction; and it may also help understand the sensory/cognitive mechanisms that mediate the cichlids' creation of large beds of shells at lake bottoms, representing large-scale modification of the environment for cichlids and many other species. The use of a continuous preference function approach (which has been useful in mate preference studies) represents an advance and its use in this paper may help spread it to research on other aspects of animal behavior. The study finds that the cichlids prefer overall larger shells with natural chirality, and also attend to other features (shell length, width, aperture) but less strongly. Size-related variation in these preferences suggests the cichlids actively adjust their refuges to their phenotype. I think these findings will be of interest to a broad audience in evolution, ecology, and behavior. My comments mainly aim to help highlight the results.

1.

I think it would be of interest to have explicit tests for sex differences in the preferences, and for a potential sex x size interaction (e.g., the male preference for shell size seems steeper than the female preference in supplementary figure 2). I appreciate that size covaries with sex (as males are larger than females). However, given an a priori reason to be interested in these questions, I would suggest not following the model selection procedure (which is better suited for cases where one does not know beforehand which variables may or may not be of interest/importance).

2.

I think the preference function trait "peak height" is difficult to interpret. It gives the highest level of response across any one variable on the x-axis, and it was included in the program PFunc in case some authors might find it of interest. However, for the authors' purpose of estimating the strength of preferences for different shell features, the trait "preference strength" is much more appropriate, as it gives an estimate of the drop in response from peak to trough along the preference function.

3.

It takes a while to get to the part that explains how the cichlids' preferences were scored (last

sentence of figure 2). This should be explained earlier --- say, at the end of the section that details the choice tasks or thereabouts.

--- In figure 2, it might also be good to explain that each data point represents the behavior/choice of one individual fish.

4.

I would suggest presenting supplementary figure 2 in the main paper, instead of the current figure 2.

Minor:

5.

Lines 362-363:

the wording that the preferences "varied in magnitude" is confusing; I think something like "varied in steepness" might be clearer.

6.

Lines 28, 406 and on:

The stimulus shells used in this experiment did not exceed the natural range observed in the cichlids' habitat. Thus, the observed open-ended preferences for larger shells do not represent preferences for supernormal stimuli. This might be the case, but there is no evidence for it in this paper.

7.

Lines 424-427:

This might be supported if the fish inspected/entered the shells and then rejected them. Do the observations have enough detail to ask this question?

Rafael L. Rodríguez

Review form: Reviewer 2

Recommendation

Accept with minor revision (please list in comments)

Scientific importance: Is the manuscript an original and important contribution to its field?

Good

General interest: Is the paper of sufficient general interest?

Good

Quality of the paper: Is the overall quality of the paper suitable?

Acceptable

Is the length of the paper justified?

No

Should the paper be seen by a specialist statistical reviewer?

Yes

Do you have any concerns about statistical analyses in this paper? If so, please specify them explicitly in your report.

No

It is a condition of publication that authors make their supporting data, code and materials available - either as supplementary material or hosted in an external repository. Please rate, if applicable, the supporting data on the following criteria.

Is it accessible?

N/A

Is it clear?

N/A

Is it adequate?

N/A

Do you have any ethical concerns with this paper?

No

Comments to the Author

Dear author,

Please find my comments in attachment (See Appendix A).

All the best.

Decision letter (RSPB-2020-0127.R0)

02-Mar-2020

Dear Dr Bose:

Your manuscript has now been peer reviewed and the reviews have been assessed by an Associate Editor. The reviewers' comments (not including confidential comments to the Editor) and the comments from the Associate Editor are included at the end of this email for your reference. As you will see, the reviewers and the Editors have raised some concerns with your manuscript and we would like to invite you to revise your manuscript to address them.

Research ethics:

Use of animals and field studies:

Please submit a copy of your revised paper within three weeks. If we do not hear from you within this time your manuscript will be rejected. If you are unable to meet this deadline please let us know as soon as possible, as we may be able to grant a short extension.

Best wishes,
 Professor Gary Carvalho
 mailto: proceedingsb@royalsociety.org

Associate Editor
 Board Member: 1
 Comments to Author:

Both reviewers were generally positive about the potential importance of the results and quality of the study design. They both found the presentation to be adequate, providing valuable conceptual and specific suggestions for improvements. In particular, as reviewer 2 notes, it may be helpful to tease-apart some sub-types of niche construction (e.g., choosing between existing options, vs. deliberate modification of the habitat vs. incidental ecosystem effects) and clarify the focus of this study (individual choice) and how that plays out for larger habitats (incidental impact on the surrounding ecosystem). In addition, it would be good to have more information regarding how the fish were housed and transported to and from the experimental set-up as well as some discussion of how the environmental conditions in this particular study may interact with the observed choice patterns and study implications.

Reviewer(s)' Comments to Author:

Referee: 1

Comments to the Author(s)

This manuscript tests for preferences for the features of the snail shells that a Lake Tanganyika cichlid fish uses as refuges. This may help understand which shell features are of importance for the cichlids' survival or reproduction; and it may also help understand the sensory/cognitive mechanisms that mediate the cichlids' creation of large beds of shells at lake bottoms, representing large-scale modification of the environment for cichlids and many other species. The use of a continuous preference function approach (which has been useful in mate preference studies) represents an advance and its use in this paper may help spread it to research on other aspects of animal behavior. The study finds that the cichlids prefer overall larger shells with natural chirality, and also attend to other features (shell length, width, aperture) but less strongly. Size-related variation in these preferences suggests the cichlids actively adjust their refuges to their phenotype. I think these findings will be of interest to a broad audience in evolution, ecology, and behavior. My comments mainly aim to help highlight the results.

1.

I think it would be of interest to have explicit tests for sex differences in the preferences, and for a potential sex x size interaction (e.g., the male preference for shell size seems steeper than the female preference in supplementary figure 2). I appreciate that size covaries with sex (as males are larger than females). However, given an a priori reason to be interested in these questions, I would suggest not following the model selection procedure (which is better suited for cases where one does not know beforehand which variables may or may not be of interest/importance).

2.

I think the preference function trait "peak height" is difficult to interpret. It gives the highest level of response across any one variable on the x-axis, and it was included in the program PFunc in case some authors might find it of interest. However, for the authors' purpose of estimating the strength of preferences for different shell features, the trait "preference strength" is much more

appropriate, as it gives an estimate of the drop in response from peak to trough along the preference function.

3.

It takes a while to get to the part that explains how the cichlids' preferences were scored (last sentence of figure 2). This should be explained earlier --- say, at the end of the section that details the choice tasks or thereabouts.

--- In figure 2, it might also be good to explain that each data point represents the behavior/choice of one individual fish.

4.

I would suggest presenting supplementary figure 2 in the main paper, instead of the current figure 2.

Minor:

5.

Lines 362-363:

the wording that the preferences "varied in magnitude" is confusing; I think something like "varied in steepness" might be clearer.

6.

Lines 28, 406 and on:

The stimulus shells used in this experiment did not exceed the natural range observed in the cichlids' habitat. Thus, the observed open-ended preferences for larger shells do not represent preferences for supernormal stimuli. This might be the case, but there is no evidence for it in this paper.

7.

Lines 424-427:

This might be supported if the fish inspected/entered the shells and then rejected them. Do the observations have enough detail to ask this question?

Rafael L. Rodríguez

Referee: 2

Comments to the Author(s)

Dear author,

Please find my comments in attachment.

All the best.

Author's Response to Decision Letter for (RSPB-2020-0127.R0)

See Appendix B.

RSPB-2020-0127.R1 (Revision)

Review form: Reviewer 1 (Rafael L. Rodriguez)

Recommendation

Accept as is

Scientific importance: Is the manuscript an original and important contribution to its field?

Excellent

General interest: Is the paper of sufficient general interest?

Excellent

Quality of the paper: Is the overall quality of the paper suitable?

Excellent

Is the length of the paper justified?

Yes

Should the paper be seen by a specialist statistical reviewer?

No

Do you have any concerns about statistical analyses in this paper? If so, please specify them explicitly in your report.

No

It is a condition of publication that authors make their supporting data, code and materials available - either as supplementary material or hosted in an external repository. Please rate, if applicable, the supporting data on the following criteria.

Is it accessible?

Yes

Is it clear?

Yes

Is it adequate?

Yes

Do you have any ethical concerns with this paper?

No

Comments to the Author

I have no further suggestions. I think this will be a trend-setting paper.

Review form: Reviewer 2

Recommendation

Accept with minor revision (please list in comments)

Scientific importance: Is the manuscript an original and important contribution to its field?

Good

General interest: Is the paper of sufficient general interest?

Good

Quality of the paper: Is the overall quality of the paper suitable?

Good

Is the length of the paper justified?

Yes

Should the paper be seen by a specialist statistical reviewer?

No

Do you have any concerns about statistical analyses in this paper? If so, please specify them explicitly in your report.

No

It is a condition of publication that authors make their supporting data, code and materials available - either as supplementary material or hosted in an external repository. Please rate, if applicable, the supporting data on the following criteria.

Is it accessible?

Yes

Is it clear?

Yes

Is it adequate?

Yes

Do you have any ethical concerns with this paper?

No

Comments to the Author

Please see attached (See Appendix C).

Decision letter (RSPB-2020-0127.R1)

06-Apr-2020

Dear Dr Bose:

Your manuscript has now been peer reviewed and the reviews have been assessed by an Associate Editor. The reviewers' comments (not including confidential comments to the Editor) and the comments from the Associate Editor are included at the end of this email for your reference. As you will see, the reviewers and the Editors have raised some concerns with your manuscript and we would like to invite you to revise your manuscript to address them.

Research ethics:

Use of animals and field studies:

All supplementary materials accompanying an accepted article will be treated as in their final form. They will be published alongside the paper on the journal website and posted on the online

figshare repository. Files on figshare will be made available approximately one week before the accompanying article so that the supplementary material can be attributed a unique DOI. Please try to submit all supplementary material as a single file.

Please submit a copy of your revised paper within three weeks. If we do not hear from you within this time your manuscript will be rejected. If you are unable to meet this deadline please let us know as soon as possible, as we may be able to grant a short extension.

Best wishes,
Professor Gary Carvalho
Editor, Proceedings B
mailto:proceedingsb@royalsociety.org

Associate Editor
Board Member: 1
Comments to Author:

The authors provide thoughtful and thorough responses to nearly all the reviewer concerns. However, Reviewer 2 notes a few areas that require further improvement. In particular, as both myself and Reviewer 2 mentioned in the original comments, environments (ie, housing parameters) are crucial to understanding potential conditionalities to the findings and need to be addressed in the main body of the MS, both the methods and discussion, at a minimum. As their response to the reviewers reveals, the fish were kept in the cylinders for the entire duration of the experiment, which introduces welfare issues that could interact with the findings. Stressed animals are biologically and behaviorally different than non-stressed animals, which can affect choices: shifting preferences or shifting strength of preference. I do not mention this issue as a fatal flaw, but rather as a biological factor that must be given due consideration in the service of building a more complete comprehension of complex systems.

Reviewer(s)' Comments to Author:
Referee: 1

Comments to the Author(s)
I have no further suggestions. I think this will be a trend-setting paper.

Referee: 2

Comments to the Author(s)
Please see attached.

Author's Response to Decision Letter for (RSPB-2020-0127.R1)

See Appendix D.

Decision letter (RSPB-2020-0127.R2)

21-Apr-2020

Dear Dr Bose

I am pleased to inform you that your manuscript entitled "Structural manipulations of a shelter resource reveal underlying preference functions in a shell-dwelling cichlid fish" has been accepted for publication in Proceedings B.

Open Access

Paper charges

Sincerely,

Professor Gary Carvalho

Associate Editor:

Board Member

Comments to Author:

Thank you for your careful attention to the remaining questions, especially regarding the efforts you took to minimize stress to the fish and explain your consideration of this aspect in the main body of the MS.

Appendix A

Review

Manipulating structural attributes of shelters reveals hidden, open-ended preference functions in a niche-constructing cichlid fish

Manuscript number: RSPB-2020-0127

Referee name: Leonor Galhardo (ISPA-IU)

GENERAL COMMENT

In general, the study needs a **minor review** in the following areas:

1. Structure of some sections
 2. Relationship niche-construction versus shell-choice
 3. In some occasions, clearly acknowledge the limitations of having performed the study without a more naturalistic social context
 4. Highlight the simple fact that animals have chosen what they are more used to in the natural environment
-

DETAILED COMMENTS

TITLE

This title is quite complex for the obtained results – We would suggest something simpler like: Preference for shell structural attributes in the shell-dwelling cichlid *N. multifasciatus*.

INTRODUCTION

General comment

The concept of niche construction is central in the introduction of this study. However, the experimental design concerns shell choices rather than analysis of niche construction related-behaviours (digging behaviour to expose the shells and sand removal from the shell's interior). Being niche construction relevant as contextual information, the focus should rather be shell use and preferences.

Line 77: Do the authors consider shell choice as part of the niche construction concept? In Lines 48-49 the concept was defined as a modification of the biotic and abiotic environments.

Line 83: Reference to the 3D-scanning should be made in Methods only.

Line 91-92: The sentence needs clarification: the choice is prior to excavation (shell's exposure) or are the authors referring to the removal of sand from inside the shell after choice?

Line 97-98: Are the larger fish belonging to the same species or not? This is a very important point for discussion in case there is a preference for large shells (smaller fish shouldn't have interest in choosing large shells).

Lines 83-109: after describing the natural behaviour of this species, this should be the place to mention which attributes of the shells have which ecological functions (Lines 126-135).

Line 113: what the authors observed was not niche-constructing behaviour but shell choice (see lines 209-210)

Line 110-125: The authors could clearly state their aim. For example:

- identify preferences for 4 different shell attributes – shell size, length, aperture and intactness;
- identify the influence of chirality in shell preferences

Following this, they could describe their predictions based on the ecological features of these animals (Line 124-125).

Line 119-123: Should be moved to methods, in a section explaining the preferences model used.

METHODS

Line 140: being this species a facultative shell breeder it would be important to state this fact early in the introduction and mention possible consequences for the shell choices.

Line 143-147: this information should be merged with the same kind of information written in the introduction.

Line 147-149: Fish features should be part of a section on 'Studied subjects' in which, some species-specific features could be added. Some information to be added here would be: Lines 184-186 and 191-195.

Still concerning the subjects, from what was stated in the introduction, brooding females or predator contexts seem to be relevant for the shell choice (Line 126-135). Given this fact, were the sexual status of males and females used identified (i.e. were they sexually mature?).

Line 149-151: should be part of the section 'shell structure'

Line 151: 'present study data', it is not a valid reference.

Line 153: Before discussing the shell structure manipulations, it would be interesting to briefly describe the equipment used and show a picture of a printed shell (e.g. what's the colour?)

Line 162: we did not have access to the supplementary materials (same for Table 1, Line 191).

Line 176: Is the extra large shell (+3SD) ecologically realistic?

Lines 195-203: This 'experimental set up' section should start with the physical features of the tanks used and anything else. Assuming that everything that has to do with fish has been already said, the experimental design follows on Lines 186-191, 203-213.

Line 195-6: why have the authors changed the aquaria dimensions? Apparently, this fact introduces an additional variable.

Line 197-199: How long did the animals stayed in the cylinder? Did you change the fish in the cylinder in each trial or did they remain the same?

Line 206: why there was a fish tested 5 times?

Line 207: Remove (JW). What was the sampling period? Did the observer have the opportunity to clearly see animals when they were hiding in shells? Later we learn that the authors have repeated trials 4 times. This should be stated here.

Line 210: 318 trials? – What happened to 2 of them (and even those related to the 5th trial above)

Line 210-211: In 10% of the trials... This should be part of the results section.

Line 216-235: Preference functions: when describing peak preference, peak height and tolerance the authors should also state the biological interpretation of these parameters (lines 294-295, lines 307-8, 323-24)

Line 234-235: results.

Line 244: why the males were not used in these tests?

Line 250: the PCAs, which explain part of Figure 2 should be better explained here in this section.

RESULTS

DISCUSSION

Line 349-358: this information is repeated; best option to make a summary of results in first paragraph.

Lines 362-364: best to go straight to actual results.

Lines 371: unpublished data? What kind of study has made these statistics, better to explain a bit more.

Line 372-375: Given this explanation, why only the large fish were keener in the choice for intactness compared with smaller fish (Line 302-303)?

Line 404: maybe these animals needed a breeding context in order to choose shells according to these features.

Line 405: these supernormal stimuli may be preferred under the context of this study because animals were alone. If there was any kind of competition/predation, would the smaller fish still choose to dig shells and be stolen/eaten inside the shell?

Line 419-430: Or a combination of both explanations.

Line 431-2: Next step can also be to simulate these naturalistic conditions in the lab.

CONCLUSION

Being relevant to have a new equipment/paradigm to study these attributes, it would be also interesting to add to this discussion the fact that animals have really chosen what is more natural (intactness and shell size), being less interested in all stimuli that is less natural (proportions of aperture and length as well as opposing chirality). Furthermore, it would be interesting to relate these results to the fact that shell intactness may be a function of the oceans acidity, which in turn are a consequence of climate change (e.g. Zhao et al 2020)

Zhao, X., Han, Y., Chen, B., Xia, B., Qu, K., & Liu, G. (2020). CO₂-driven ocean acidification weakens mussel shell defense capacity and induces global molecular compensatory responses. *Chemosphere*, 243, 125415.

Appendix B

Centre for the Advanced Study
of Collective Behaviour

Max Planck Institute
of Animal Behavior

Editor-In-Chief

Dr. Spencer Barrett
University of Toronto

Editor

Dr. Gary Carvalho
University of Bangor

March 16, 2020

Dear Drs. Barrett and Carvalho,

We are grateful to you and the two referees for facilitating the review of our manuscript (RSPB-2020-0127). The reviews were thorough, insightful and provided welcome constructive feedback. We have accordingly revised the manuscript. In brief, following the suggestions of reviewer 1, we have addressed the issue of the strong correlation between sex and body size and explained how we dealt with it statistically. We have also taken their suggestion of analyzing “preference strength” rather than “peak height”, which is a more appropriate preference function trait for our study. Following the suggestions of reviewer 2, we have made considerable changes to the text to better explain how shell choice in our model study species is embedded in the wider concept of niche construction. We have also added more information about experimental methodology. Below, we have responded to each of the reviewer’s comments leaving their comments in plain text and writing our responses in bold text.

We thank you again for your assistance and look forward to hearing from you at your earliest convenience.

Sincerely,

Aneesh P. H. Bose (corresponding author)

Associate Editor

Board Member: 1

Comments to Author:

Both reviewers were generally positive about the potential importance of the results and quality of the study design. They both found the presentation to be adequate, providing valuable conceptual and specific suggestions for improvements. In particular, as reviewer 2 notes, it may be helpful to tease-apart some sub-types of niche construction (e.g., choosing between existing options, vs. deliberate modification of the habitat vs. incidental ecosystem effects) and clarify the focus of this study (individual choice) and how that plays out for larger habitats (incidental impact on the surrounding ecosystem). In addition, it would be good to have more information regarding how the fish were housed and transported to and from the experimental set-up as well as some discussion of how the environmental conditions in this particular study may interact with the observed choice patterns and study implications.

> We thank you and the two reviewers for the efforts made to improve our manuscript. Please see below for our point-by-point responses. Please note that all line numbers listed below are made in reference to the revised text that hides all visible track changes (i.e. 'Simple Markup' or 'No Markup' in Microsoft Word).

Reviewer(s)' Comments to Author:

Referee: 1

Comments to the Author(s)

This manuscript tests for preferences for the features of the snail shells that a Lake Tanganyika cichlid fish uses as refuges. This may help understand which shell features are of importance for the cichlids' survival or reproduction; and it may also help understand the sensory/cognitive mechanisms that mediate the cichlids' creation of large beds of shells at lake bottoms, representing large-scale modification of the environment for cichlids and many other species. The use of a continuous preference function approach (which has been useful in mate preference studies) represents an advance and its use in this paper may help spread it to research on other aspects of animal behavior. The study finds that the cichlids prefer overall larger shells with natural chirality, and also attend to other features (shell length, width, aperture) but less strongly. Size-related variation in these preferences suggests the cichlids actively adjust their refuges to their phenotype. I think these findings will be of interest to a broad audience in evolution, ecology, and behavior. My comments mainly aim to help highlight the results.

> Thank you for the positive assessment of our work. We hope that we have adequately addressed all concerns below.

1. I think it would be of interest to have explicit tests for sex differences in the preferences, and for a potential sex x size interaction (e.g., the male preference for shell size seems steeper than the female preference in supplementary figure 2). I appreciate that size covaries with sex (as males are larger

than females). However, given an a priori reason to be interested in these questions, I would suggest not following the model selection procedure (which is better suited for cases where one does not know beforehand which variables may or may not be of interest/importance).

> We completely agree that testing for sex differences would be of biological interest. Unfortunately, however, we think that the correlation between sex and body size in our dataset is too strong to allow us to make reliable inferences about sex and body size separately. We investigated whether we could reliably include both sex and body size into our models, by calculating variance inflation factors (VIFs) to quantify the severity of multicollinearity arising due to the correlation between sex and body size. These inflation factors were deemed to be too high (i.e. sometimes above 5) even after taking measures to reduce multicollinearity (e.g. standardizing). We recognize that an arbitrary threshold is sometimes an ineffectual way of diagnosing such problems (see O'Brien et al. 2007). However, given our sample sizes and a lack of other model features that would reduce standard error inflation, we believe that any inferences made from estimates given by models that contain both sex and body size together would be tenuous. Further evidence of multicollinearity in our data comes from inspecting output from models that contain just sex, just body size, or both. When models contained sex or body size on their own, they were both strongly significant predictors of our response variables. However, neither were significant, due to inflated standard errors, when they were both included in the

models together.

Figure illustrating the strong sex-size relationship across our experimental groups.

We have now added more information to the main text of the manuscript, in order to better convey the strong relationship between sex and body size, and how we handled this. See lines 279-291.

O'brien, R. M. (2007). A caution regarding rules of thumb for variance inflation factors. Quality & quantity, 41(5), 673-690.

2. I think the preference function trait "peak height" is difficult to interpret. It gives the highest level of response across any one variable on the x-axis, and it was included in the program PFunc in case

some authors might find it of interest. However, for the authors' purpose of estimating the strength of preferences for different shell features, the trait "preference strength" is much more appropriate, as it gives an estimate of the drop in response from peak to trough along the preference function.

> Thank you for pointing this out. "Preference strength" is indeed a more useful trait. We have updated our analyses, replacing "peak height" with "preference strength". Our results have not qualitatively changed. See lines 228-230, 323-338, 646 and new Figure 3.

3. It takes a while to get to the part that explains how the cichlids' preferences were scored (last sentence of figure 2). This should be explained earlier --- say, at the end of the section that details the choice tasks or thereabouts.

> We have now added a line stating the fish's shell choices were used to calculate their preference functions. See lines 210-212.

--- In figure 2, it might also be good to explain that each data point represents the behavior/choice of one individual fish.

> We have modified the last sentence of the figure 2 caption to better convey this point. See lines 643-644.

4. I would suggest presenting supplementary figure 2 in the main paper, instead of the current figure 2.

> We appreciate the suggestion; however, we feel that the current figure 2 conveys the appropriate information given the analyses presented in the main text. The main analyses do not (and cannot) explicitly differentiate between the sexes and so presenting the sexes separately in plots represents a disconnect between the analyses and the data visualizations.

Minor:

5. Lines 362-363:

the wording that the preferences "varied in magnitude" is confusing; I think something like "varied in steepness" might be clearer.

> We have now reworded this section to be more specific. See lines 384-385.

6. Lines 28, 406 and on:

The stimulus shells used in this experiment did not exceed the natural range observed in the cichlids' habitat. Thus, the observed open-ended preferences for larger shells do not represent preferences for supernormal stimuli. This might be the case, but there is no evidence for it in this paper.

> The largest or most exaggerated shell replicas did, in fact, exceed the natural range observed in the wild (based on our sample of 113 wild-collected shells). However, the largest replicas only exceeded the limits of the natural range by a very small margin. For example, the largest shell

replicas that we printed were 60.6 mm in length, but the longest natural shell from the wild was 60.0 mm. Therefore, it is correct to point out that this is not a critical test of supernormal stimuli, but rather a test for features that we exaggerated to their most extreme, but still ecologically-valid, forms. We have therefore tempered our wording on lines 29 and 430-433, acknowledging this.

7. Lines 424-427:

This might be supported if the fish inspected/entered the shells and then rejected them. Do the observations have enough detail to ask this question?

> That is an excellent suggestion, unfortunately, we do not have the behavioural data from these chirality trials to distinguish between an aversion to novel stimuli and an evolutionary lateralization.

Rafael L. Rodríguez

Referee 2:

GENERAL COMMENT

In general, the study needs a minor review in the following areas:

1. Structure of some sections
2. Relationship niche-construction versus shell-choice
3. In some occasions, clearly acknowledge the limitations of having performed the study without a more naturalistic social context
4. Highlight the simple fact that animals have chosen what they are more used to in the natural environment

> Thank you for the detailed comments on our manuscript. We believe that by addressing them (below), the manuscript is much improved.

DETAILED COMMENTS

TITLE

1. This title is quite complex for the obtained results – We would suggest something simpler like: Preference for shell structural attributes in the shell-dwelling cichlid *N. multifasciatus*.

> We agree that the previous title was rather wordy. Our new title is more concise. See lines 1-2.

INTRODUCTION

2. General comment: The concept of niche construction is central in the introduction of this study. However, the experimental design concerns shell choices rather than analysis of niche construction related-behaviours (digging behaviour to expose the shells and sand removal from the shell's interior). Being niche construction relevant as contextual information, the focus should rather be shell use and preferences.

> We have reworded the first two paragraphs of the introduction to place more emphasis on resource choice, and placed it into the broader context of niche construction. See lines 43-81.

3. Line 77: Do the authors consider shell choice as part of the niche construction concept? In Lines 48-49 the concept was defined as a modification of the biotic and abiotic environments.

> This section has now been reworded as per the above comment. In the *Neolamprologus multifasciatus* system, we do consider shell choice a component of niche construction. This is because the choice of which shells to excavate and to maintain is an important part of the generation of shell beds on the lake floor, which support great biodiversity. We make this point more clearly on lines 91-95 and 114-117.

4. Line 83: Reference to the 3D-scanning should be made in Methods only.

> We have now removed mention of 3D-scanning from this sentence. See lines 82-83.

5. Line 91-92: The sentence needs clarification: the choice is prior to excavation (shell's exposure) or are the authors referring to the removal of sand from inside the shell after choice?

> We have clarified this sentence. Shell choice is the first step in the niche construction process. Shells must first be chosen, and then maintained to be free of sediment. See lines 91-95.

6. Line 97-98: Are the larger fish belonging to the same species or not? This is a very important point for discussion in case there is a preference for large shells (smaller fish shouldn't have interest in choosing large shells).

> Here, we are referring to larger heterospecifics. This has now been made clear on lines 96-100.

7. Lines 83-109: after describing the natural behaviour of this species, this should be the place to mention which attributes of the shells have which ecological functions (Lines 126-135).

> We have now made rearrangements to the introduction, as per reviewer 2's suggestions. See lines 101-110.

8. Line 113: what the authors observed was not niche-constructing behaviour but shell choice (see lines 209-210)

> We have changed our wording here (and elsewhere throughout the MS) to better explain that our main observations were of shell choice, but that shell choice is an integral component of niche construction in our study species. See lines 114-117 and 372-374.

9. Line 110-125: The authors could clearly state their aim. For example:

- identify preferences for 4 different shell attributes – shell size, length, aperture and intactness;
- identify the influence of chirality in shell preferences

Following this, they could describe their predictions based on the ecological features of these animals (Line 124-125).

> Thank you, we have now made the suggested rearrangements. See lines 113-134.

10. Line 119-123: Should be moved to methods, in a section explaining the preferences model used.

> We respectfully disagree. Outlining the different possible preference function shapes (i.e. flat vs. curved, closed vs. open) is important for presenting our predictions to the reader. I.e. if an attribute is important for the animal's decision making, then we expect preference functions that are curved, and the shape of this curve should point towards which attribute values are most preferred.

METHODS

11. Line 140: being this species a facultative shell breeder it would be important to state this fact early in the introduction and mention possible consequences for the shell choices.

> We now explain that *N. multifasciatus* is a facultative shell breeder, but they always use shells in regions of the lake where shells are present (i.e. shell beds). See lines 86-87.

12. Line 143-147: this information should be merged with the same kind of information written in the introduction.

> We prefer to keep the natural history information of *N. multifasciatus* to a minimum in the introduction, and leave this information in the "study species" section of the methods. Our introduction has also been reworded extensively as per the reviewers' comments, and this has already led to a new balance of natural history information presented in the introduction versus the methods sections.

13. Line 147-149: Fish features should be part of a section on 'Studied subjects' in which, some species-specific features could be added. Some information to be added here would be: Lines 184-186 and 191-195.

> The "study species" section of the methods is meant to provide the reader with extra relevant contextual information about the natural history of our focal species. This section is not meant to give the reader details about the specific body sizes and sex breakdown of the sample of individuals that we used in our lab experiment. That information is reserved for later sections of the methods.

14. Still concerning the subjects, from what was stated in the introduction, brooding females or predator contexts seem to be relevant for the shell choice (Line 126-135). Given this fact, were the sexual status of males and females used identified (i.e. were they sexually mature?).

> Yes, all fish used in our experiment were adults and sexually mature. This has now been clarified on lines 183-186.

15. Line 149-151: should be part of the section 'shell structure'

> This information has now been moved to the 'shell structure' section. See lines 151-153.

16. Line 151: 'present study data', it is not a valid reference.

> This reference has now been removed. See line 153.

17. Line 153: Before discussing the shell structure manipulations, it would be interesting to briefly describe the equipment used and show a picture of a printed shell (e.g. what's the colour?)

> We have now modified figure 1 to include a panel showing a printed shell. Extra information on the equipment can be found in the supplementary materials. See figure 1C and lines 631-632.

18. Line 162: we did not have access to the supplementary materials (same for Table 1, Line 191).

> We are sorry that you did not receive the supplementary information. They were uploaded as a separate file. We have now also appended the supplementary materials section to the end of this file. See below.

19. Line 176: Is the extra large shell (+3SD) ecologically realistic?

> The extra large shell is at the extreme limits of what can be found in the wild. Please see our response above to reviewer 1's comment 6. Also see lines 429-433.

20. Lines 195-203: This 'experimental set up' section should start with the physical features of the tanks used and anything else. Assuming that everything that has to do with fish has been already said, the experimental design follows on Lines 186-191, 203-213.

> The tank dimensions are presented where they are because we first needed to introduce the various choice tasks performed in our experiment. Once that was explained, we could present tank dimensions, which were slightly different in one of the choice tasks. Also, please see our response to the comment below.

21. Line 195-6: why have the authors changed the aquaria dimensions? Apparently, this fact introduces an additional variable.

> Due to logistical reasons and limited experimental aquaria, we were forced to use several aquaria of slightly different dimensions for one of the choice tasks. However, regardless of the dimensions used, all tanks provided ample space for the shell-choosing fish. Furthermore, tank dimensions did not vary within choice tasks and so did not introduce an additional variable that would require statistical consideration.

22. Line 197-199: How long did the animals stay in the cylinder? Did you change the fish in the cylinder in each trial or did they remain the same?

> The stimulus groups remained the same across trials and they remained in the cylinder for the entire duration of the experiment. More information on animal housing conditions can be found in the supplementary materials section. See it appended below.

32. Line 206: why there was a fish tested 5 times?

> This was done because there was, at one point, the time and opportunity to test a fish in an additional trial, and this fish was chosen haphazardly from the sample. However, this does not detract from the preference function data that we obtained from this fish; in fact, it allows for a better resolution of the shape of the preference curve.

33. Line 207: Remove (JW). What was the sampling period? Did the observer have the opportunity to clearly see animals when they were hiding in shells? Later we learn that the authors have repeated trials 4 times. This should be stated here.

> We have now removed "(JW)". The sampling period is now stated in the supplementary materials section. Yes, the observer was able to see which shell was occupied during their spot checks. We actually mention that the trials were repeated 4 times in the sentence just prior to the one mentioning JW.

34. Line 210: 318 trials? – What happened to 2 of them (and even those related to the 5th trial above)

> Thank you for pointing out this discrepancy. We realized that there was some information missing here. There were in fact 3 fish (out of the 80) that were tested only 3 times each, and 1 fish (out of the 80) that was tested 5 times. The total number of trials sums to 318. See lines 204-207.

35. Line 210-211: In 10% of the trials... This should be part of the results section.

> We have opted to move this information to lines 238-240 instead, where it is better suited to inform how the preference functions were calculated.

36. Line 216-235: Preference functions: when describing peak preference, peak height and tolerance the authors should also state the biological interpretation of these parameters (lines 294-295, lines 307-8, 323-24)

> We have now included a sentence which comments on the biological importance of these preference function traits. See lines 233-235.

37. Line 234-235: results.

> We prefer to keep this information where it is, because it is relevant for the reader to understand the methodology, in particular which trials were included in the statistical analyses.

38. Line 244: why the males were not used in these tests?

> We recognize that the male side of the chirality story was not told in the original version of our manuscript. As such, we have run additional trials since receiving these reviews and we now include male tests in the revised manuscript. The male data do not deviate significantly from the female data, and our results have not changed qualitatively. *N. multifasciatus* individuals still express a significant preference for shells displaying the normal, wildtype, chirality. See lines 250-252 and 361-366.

39. Line 250: the PCAs, which explain part of Figure 2 should be better explained here in this section.

> The PCA was not meant to be part of the statistical analyses, as would be implied by putting a description of it here. It was simply a tool that we used to create a realistic density plot for the natural population distribution of 'shell size' against which we could plot our group-level preference functions.

RESULTS

DISCUSSION

40. Line 349-358: this information is repeated; best option to make a summary of results in first paragraph.

> We would like to keep this short paragraph for stylistic reasons. It helps to remind the reader of the broader context around our results.

41. Lines 362-364: best to go straight to actual results.

> We have now shortened the first few lines of this paragraph in order to get to the results faster. See lines 380-384.

42. Lines 371: unpublished data? What kind of study has made these statistics, better to explain a bit more.

> These statistics came from field surveys conducted in September 2019 in which all the shells within 24 *N. multifasciatus* were collected, counted, and inspected for intactness. This has now been explained in more detail on lines 394-397.

43. Line 372-375: Given this explanation, why only the large fish were keener in the choice for intactness compared with smaller fish (Line 302-303)?

> We apologize for the misunderstanding here. Larger bodied fish had stronger preferences for exaggerated or intact shells than smaller bodied fish, however, *all fish* still valued exaggerated and intact shells. This has now been clarified on lines 319-321.

44. Line 404: maybe these animals needed a breeding context in order to choose shells according to these features.

> We completely agree with this point. In fact, we raise this issue in a later discussion paragraph where we talk about bringing our experimental paradigm into the field, or more natural contexts, where animals face a fuller range of trade-offs. See lines 459-470.

45. Line 405: these supernormal stimuli may be preferred under the context of this study because animals were alone. If there was any kind of competition/predation, would the smaller fish still choose to dig shells and be stolen/eaten inside the shell?

> Unfortunately, we cannot answer this question with the available data as it is outside the scope of our study. However, this is a point we raise in a later discussion paragraph on conducting this work in the field. See lines 459-470.

46. Line 419-430: Or a combination of both explanations.

> We have now written that these explanations are non-mutually exclusive. See line 449.

47. Line 431-2: Next step can also be to simulate these naturalistic conditions in the lab.

> We have incorporated this into lines 459-461.

CONCLUSION

48. Being relevant to have a new equipment/paradigm to study these attributes, it would be also interesting to add to this discussion the fact that animals have really chosen what is more natural (intactness and shell size), being less interested in all stimuli that is less natural (proportions of aperture and length as well as opposing chirality).

> We now mention that the most natural shell forms were the ones that were most attractive to the fish. See lines 380-384.

49. Furthermore, it would be interesting to relate these results to the fact that shell intactness may be a function of the oceans acidity, which in turn are a consequence of climate change (e.g. Zhao et al 2020)

Zhao, X., Han, Y., Chen, B., Xia, B., Qu, K., & Liu, G. (2020). CO₂-driven ocean acidification weakens mussel shell defense capacity and induces global molecular compensatory responses. *Chemosphere*, 243, 125415.

> While ocean acidification is indeed a rising problem and one that will likely impact the production of calcium carbonate shells, this lies a little beyond the scope of our current manuscript. First, Lake Tanganyika is a freshwater environment and climate change's impacts on lake pH is currently unknown. Second, we believe that a discussion on how climate change could be projected to reduce shell intactness forces too much emphasis on the shells themselves. Instead, we would like to keep the broader focus on how our paradigm can be used across taxa and across contexts to study animal resource choice decisions.

Appendix C

Dear authors,

I acknowledge that you have thoroughly revised the manuscript in light of previous comments. I carefully read your explanations and decisions and agree with them. Below are just some remaining minor notes I send you for consideration. I wish you all the success!

Leonor Galhardo

Comment 17.

In the discussion, the authors could acknowledge the fact that the shell colour was not a controlled variable and that future studies could analyse colour preferences as well as degree of mimetism against the chosen substrate.

Comment 21.

This same explanation should be made in the text just for the reader to understand the reasons behind the different sizes.

Comment 22.

The fact that these fish stayed in the cylinder for the entire period of the experiment (3,5 months) is, from the welfare's perspective, a weakness of this experimental design. These fish could have spent some interspersed time in their home tanks, or be replaced by other individuals. Therefore, the authors could provide an explanation on why they made this decision and make clear that a different approach would be difficult for them.

Comment 32.

Still, it seems to me a questionable approach. Preferably, number of trials should always be the same.

Comment 34.

So, in my view, the total sample analysed should include only fish that performed 4 trials.

Comment 38.

It was very efficient of you!

Comment 39.

The statistical analysis includes the preliminary exploration of data (e.g. PCA) and the inference analysis. The PCA was a good approach of you, but still requires an explanation for better understanding of this exploratory step. This should be done before the inference analysis.

Comment 42.

Maybe then rather use 'personal observation'

Comment 43.

Thank you very much for your explanation and sorry for the misunderstanding. In this case, it would be interesting to mention why you think the small fish (females) would prefer exaggerated attributes when this choice can turn against them?

Appendix D

Associate Editor

Board Member: 1

Comments to Author:

The authors provide thoughtful and thorough responses to nearly all the reviewer concerns. However, Reviewer 2 notes a few areas that require further improvement. In particular, as both myself and Reviewer 2 mentioned in the original comments, environments (ie, housing parameters) are crucial to understanding potential conditionalities to the findings and need to be addressed in the main body of the MS, both the methods and discussion, at a minimum. As their response to the reviewers reveals, the fish were kept in the cylinders for the entire duration of the experiment, which introduces welfare issues that could interact with the findings. Stressed animals are biologically and behaviorally different than non-stressed animals, which can affect choices: shifting preferences or shifting strength of preference. I do not mention this issue as a fatal flaw, but rather as a biological factor that must be given due consideration in the service of building a more complete comprehension of complex systems.

> Thank you for your efforts in reviewing our manuscript. Please see below for our point-by-point responses to the last few comments. With respect to the comment on housing parameters, we now explain that the conditions we implemented during the experiment were carefully chosen to *minimize* stress. None of the animals used in our experiment experienced unnaturally cramped or isolated conditions, and we attempted to give the stimulus groups within the cylinders an environment that mimicked what they would experience in the wild. All animals displayed their natural repertoire of behaviours over the course of our work.

Reviewer(s)' Comments to Author:

Referee: 1

Comments to the Author(s)

I have no further suggestions. I think this will be a trend-setting paper.

> Thank you for the positive assessment of our manuscript!

Referee: 2

Dear authors,

I acknowledge that you have thoroughly revised the manuscript in light of previous comments. I carefully read your explanations and decisions and agree with them. Below are just some remaining minor notes I send you for consideration. I wish you all the success!

Leonor Galhardo

> Thank you and we hope that we have now adequately addressed your final comments.

Comment 17.

In the discussion, the authors could acknowledge the fact that the shell colour was not a controlled variable and that future studies could analyse colour preferences as well as degree of mimetism against the chosen substrate.

> We agree that varying shell colour (or surface texture, or any number of other attributes) would be very interesting. We have raised this point on lines 422-424. The white colour of our shell replicas, however, was neither unintentional nor without forethought. This is because in the wild, relatively 'new' shells (shortly after the snail dies and leaves the empty shell behind) are often white. The shells then take on additional colour and algal growth as they age.

Comment 21.

This same explanation should be made in the text just for the reader to understand the reasons behind the different sizes.

> We have now added this explanation to the text. See lines 199-200.

Comment 22.

The fact that these fish stayed in the cylinder for the entire period of the experiment (3,5 months) is, from the welfare's perspective, a weakness of this experimental design. These fish could have spent some interspersed time in their home tanks, or be replaced by other individuals. Therefore, the authors could provide an explanation on why they made this decision and make clear that a different approach would be difficult for them.

> We chose this approach because it was the one that actually minimized stress on all the animals. It was only the stimulus group that remained within the cylinder for the duration of the experiment. The shell-choosing fish was free to move about the entire aquarium. However, in the wild, *Neolamprologus multifasciatus* individuals do not venture far from their shells or their territory. In fact, these fish spend the majority of their lifetimes in an area on the lake floor not much larger than what the cylinder provided. Thus, the cylinder does not represent unnaturally cramped conditions for the fish. Continuous removal and reintroduction of the stimulus group would have caused undue stress due to handling and the constant need to re-establish their territory and social relationships once placed back into the cylinder. We add information to this effect on lines 205-207 and in the "fish housing conditions" paragraph in Supplementary Materials.

Comment 32.

Still, it seems to me a questionable approach. Preferably, number of trials should always be the same.

> Please see the comment below for our response.

Comment 34.

So, in my view, the total sample analysed should include only fish that performed 4 trials.

> We chose to include all fish that we tested in our analyses. Including all the fish, even though a very small proportion of them were tested once more or once less, does not introduce any consistent bias in our data. Omitting these fish from our analyses also does not qualitatively change our results or conclusions. We now say so on lines 212-213.

Comment 38.

It was very efficient of you!

Comment 39.

The statistical analysis includes the preliminary exploration of data (e.g. PCA) and the inference analysis. The PCA was a good approach of you, but still requires an explanation for better understanding of this exploratory step. This should be done before the inference analysis.

> We have now introduced the exploratory PCA at an earlier point in the main-text of the manuscript where we think it is suitable. See lines 157-159.

Comment 42.

Maybe then rather use 'personal observation'

> We have changed our wording to 'personal observation'. See line 402.

Comment 43.

Thank you very much for your explanation and sorry for the misunderstanding. In this case, it would be interesting to mention why you think the small fish (females) would prefer exaggerated attributes when this choice can turn against them?

> Our interpretation of this is that smaller fish (females) also prefer exaggerated shells (though their preferences are not as strong). Importantly, however, small fish in the wild may not always be able to secure their most preferred shell type because of limited shell availability and the fact that larger fish will monopolize the most valuable resources. This is interesting because it suggests that natural surveys of fish in the wild will not reveal true preferences because true preferences become masked by resource competition and limited choices. See lines 435-439.